# Differences in Ectopic Pregnancy Rates between Fresh and Frozen Embryo Transfer after In Vitro Fertilization: A Large Retrospective Study

**DOI:** 10.3390/jcm11123386

**Published:** 2022-06-13

**Authors:** Zhijie Hu, Danjun Li, Qiuju Chen, Weiran Chai, Qifeng Lyu, Renfei Cai, Yanping Kuang, Xuefeng Lu

**Affiliations:** Department of Assisted Reproduction, Shanghai Ninth People’s Hospital, Shanghai Jiaotong University School of Medicine, 639 Zhizaoju Rd, Shanghai 200011, China; huzhijie0204@163.com (Z.H.); 15726425652@163.com (D.L.); chenqj75@126.com (Q.C.); chaiwrcn@163.com (W.C.); lyuqifeng@126.com (Q.L.); cairenfei070@sina.com (R.C.)

**Keywords:** ectopic pregnancy, fresh embryo transfer, frozen embryo transfer, ovarian stimulation

## Abstract

Ectopic pregnancy (EP) is increasingly found in women treated with in vitro fertilization and embryo transfer (IVF–ET). With the development of the freeze-all policy in reproductive medicine, it is controversial whether frozen embryo transfer (FET) could reduce the rate of EP. In this single-center, large-sample retrospective study, we analyzed 16,048 human chorionic gonadotrophin (hCG)-positive patients who underwent fresh embryo transfer (ET) or FET cycles between January 2013 and March 2022. Throughout the study, the total EP rate was 2.09% (336/16,048), 2.16% (82/3803) in the ET group, and 2.07% (254/12,245) in the FET group. After adjustment for age, infertility causes, and other confounding factors, logistic regression results showed no statistical difference in EP rates between FET and ET groups (odds ratio (OR) 0.93 (0.71–1.22), *p* > 0.05). However, among the 3808 patients who underwent fresh ET cycles, the OR for EP was significantly lower in the long agonist protocol group than in the gonadotropin-releasing hormone antagonist (GnRH-ant) protocol group (OR 0.45 (0.22–0.93), *p* < 0.05). Through a large retrospective study, we demonstrated a slightly lower EP rate in FET cycles than in fresh ET cycles, but there was no significant difference. The long agonist protocol in ET cycles had a significantly lower risk of EP than the GnRH-ant protocol.

## 1. Introduction

Ectopic pregnancy (EP), an extrauterine implantation of the embryo, is a serious complication of in vitro fertilization and embryo transfer (IVF–ET). The incidence of EP after IVF–ET has been reported to be around 2–8%, higher than that after natural conception [1,2,3,4]. Several studies have shown that the risk factors associated with high EP rate after IVF–ET include tubal infertility, pelvic inflammatory disease, history of tubal surgery, previous EP, cigarette smoking, endometriosis etc. [4,5]. In addition to the maternal-related factors, risk factors associated with the IVF–ET technique have also been elucidated to further increase the risk of EP. Previous studies have shown that EP is related with high levels of estrogen, which lead to increased uterine contractions [6,7], and that it is also related to high levels of progesterone, leading to ciliary dysfunction during assisted reproductive technology (ART) [8].

In recent years, many researchers have suggested that the type of embryo transferred (fresh or frozen embryo transfer) might be related to EP rate. Some studies have claimed that fresh embryo transfer (ET) occurred in a supraphysiological hormonal environment that affected uterine contractility [9] or endometrial tolerance [10], whereas frozen embryo transfer (FET) occurred in a uterine environment more similar to natural conception; thus, the incidence of EP was significantly lower with FET than with fresh ET [11,12,13]. Osamu Ishihara‘s and Huang’s studies both showed that the higher EP rate after fresh ET was related to the negative effect of ovarian stimulation on endometrial receptivity, but the majority of FET used natural cycle preparation [11,12].

With the development of the freeze-all policy in reproductive medicine, it is controversial whether FET could reduce the rate of EP. Some studies have reported a higher EP rate in FET compared with ET, possibly due to defects in embryo quality after the freeze–thaw process [14]. Moreover, the developmental delay of thawed embryos might provide additional opportunities for embryos to migrate to fallopian tube before implantation in the uterus [15]. Ultimately, some studies have suggested that the EP rate between FET and ET was comparable [16,17,18].

However, many previous studies did not include important risk factors associated with EP as confounding factors to adjust the results, especially the lack of data on EP history, thereby limiting the accuracy of the findings. In addition, ET cycles were performed in a single protocol, and the effect of different ovarian stimulation protocols on the EP rate was not analyzed, thereby limiting the generalizability of the findings.

In this study, we designed a single-center, large-sample, retrospective study to analyze EP rates in fresh ET or FET cycles, and also under different ovarian stimulation protocols, integrating risk factors associated with EP to ensure the accuracy and generalizability of the results. We hypothesized that the EP rate might be associated with the type of embryo transferred (fresh or frozen embryo transfer).

## 2. Materials and Methods

### 2.1. Study Design and Participants

This retrospective study was conducted from January 2013 to March 2022 at the Department of Assisted Reproduction of the Ninth People’s Hospital of Shanghai JiaoTong University School of Medicine. We included 16,050 human chorionic gonadotrophin (hCG) -positive patients from fresh ET cycles and FET cycles. We excluded 2 patients who were lost to follow-up. Finally, 16,048 IVF–ET cycles were included, which were divided into fresh ET group and FET group. According to different ovarian stimulation protocols, the ET group was further divided into a gonadotropin-releasing hormone antagonist (GnRH-ant) protocol group, a prolonged agonist protocol group, a short agonist protocol group, and a long agonist protocol group. We compared the EP rate between each group. The study protocol was approved by the ethics committee of the Ninth Hospital. Figure 1 shows the flow chart of this study (Figure 1).

### 2.2. Ovarian Stimulation, Monitoring and Oocyte Retrieval Operation

Controlled ovarian stimulation was performed under a GnRH-ant protocol, a long agonist protocol, a short agonist protocol, or a prolonged agonist protocol, depending on the patient’s condition.

Under the long agonist protocol, a long-acting gonadotropin-releasing hormone agonist (GnRH-α) (leuprorelin acetate, 3.75 mg) was administered on day 2 to day 5 of the cycle. If downregulation was quantified 35 days later, human menopausal gonadotropin (hMG) (150–225 IU/day) would be given until the trigger day. In the prolonged agonist protocol, extended-release GnRH-α 1.5–2.0 mg was administered on the twentieth day of the menstrual cycle (MC), and reinjected 30 days later. Ovulation was initiated with hMG if the estrogen level was lower than 40 pg/mL and the sinus follicle was of appropriate size (4–6 mm) 12 days later. Under the short agonist protocol, triptorelin 0.1 mg/day was administered from MC2, and hMG (150–225 IU/day) was injected from MC3. Under the GnRH-ant protocol, hMG (150–225 IU/day) was given from MC3. When the dominant follicle reached approximately 14 mm in diameter five days later, the gonadotropin-releasing hormone (GnRH) antagonist, certrotide (0.125–0.25 mg/day), was administered until trigger day.

Ovulation was triggered by 5000 IU hCG when three dominant follicles reached at least 18 mm in diameter or when one dominant follicle reached 20 mm in diameter, except for the GnRH-ant protocol, which triggered ovulation with 0.1 mg triptorelin and 5000 IU hCG. The timing of oocyte retrieval depended on the protocol.

All follicles ≥ 10 mm in diameter were aspirated using a double-lumen retrieval needle under vaginal ultrasound guidance.

### 2.3. Insemination, and Embryo Culture

The retrieved oocytes were subjected to in vitro fertilization (IVF) or intracytoplasmic sperm injection (ICSI), according to routine laboratory procedures. The quality of embryos was observed on the third day, and graded according to Cummins standards [19], with grade I and II embryos (6 cells and above) defined as high quality embryos. High quality embryos were frozen through vitrification, while the rest were cultured until day 5/6, and the good quality blastocysts were frozen for subsequent FET cycles.

### 2.4. Fresh Embryo Transfer

On the third day after oocyte retrieval, 1–2 high quality embryos were selected to be transferred into the uterine cavity. Luteal support was started with oral progesterone (Duphaston) 40 mg/day and vaginal utrogestan 0.4 g/day on the day of oocyte retrieval. The dose would be adjusted according to the pregnancy test results 14 days after embryo transfer.

### 2.5. Endometrial Preparation and Frozen Embryo Transfer

Endometrial preparation and the FET were performed as described in our previous study [20]. The endometrial preparation was performed under the natural cycle, the ovulation cycle, or the hormone replacement cycle, according to the patient’s condition. The natural cycle was used for patients with regular menstrual cycles, and the ovulation cycle was indicated for patients with previous irregular menstrual cycles. Hormone replacement therapy (HRT) was used if the patient had recurrent thin endometria on the natural or ovulation cycle. For all three protocols, embryos aged 3 or 5–6 days old were thawed for transfer on the third day of endometrium translation.

### 2.6. Outcome Measures

The primary outcome was the EP rate. The second outcome was the heterotopic pregnancy rate. Serum β-hCG levels were measured 14 days after embryo transfer. If β-hCG was positive, a vaginal ultrasound was performed 35 days after embryo transfer.

EP was defined as a gestational sac observed via ultrasound outside of the uterine cavity. Heterotopic pregnancy was defined as the coexistence of a clinical intrauterine pregnancy and an EP. EP rate was calculated as the number of EPs per 100 positive β-hCG tests after IVF–ET. The heterotopic pregnancy rate was calculated as the number of heterotopic pregnancies per 100 positive β-hCG tests after IVF–ET.

### 2.7. Statistical Analysis

All data were analyzed using the Statistical Package for Social Sciences version 26.0 (SPSS; IBM, Armonk, NY, USA). For continuous variables, the normality was measured by the Kolmogorov–Smirnov test and Q-Q plots. If the data were normally distributed, they were shown as the mean ± standard deviation, or else they were shown as median (first quartile–third quartile). Student’s *t*-test (normal distribution) or a Mann–Whitney U test (no normal distribution) was performed on the continuous variables; a chi-squared test was performed on categorical variables. Categorical variables that do not satisfy the chi-squared test were tested by Fisher’s precision probability test. We used binary logistic regression to investigate the independent effect of embryo transfer type (fresh ET versus FET) and different ovarian stimulation protocols on the odds ratio (OR) of EP, as well as the regression model adjusted for age, body mass index (BMI), infertility duration, infertility causes, history of ectopic pregnancy, nature cycle, number of embryos transferred, stage of embryos transferred, and endometrial thickness. The logistic regression results were shown as OR (95% confidence interval). *p* < 0.05 was considered statistically significant.

## 3. Results

### 3.1. Patient and Cycle Characteristics

A total of 16,050 hCG-positive IVF–ET cycles occurred during the study period, and 2 patients who were lost to follow-up after clinical pregnancy were excluded, resulting in the inclusion of 16,048 IVF–ET cycles, including a total of 3803 cycles in the fresh ET group and 12,245 cycles in the FET group (Figure 1). We found significant differences between the two groups in terms of age, BMI, infertility duration, infertility diagnosis, history of EP, number of embryos transferred, stage of embryos transferred, and endometrial thickness. We included all of these characteristics as confounding factors in the regression equation for the comparison of EP rates (Table 1).

### 3.2. Ectopic Pregnancy Outcome between FET and ET Group

Throughout the study, 336 EPs occurred, with 82 in the ET group and 254 in the FET group. The total EP rate was 2.09%. The EP rate was 2.16% in the ET group and 2.07% in the FET group. A total of 47 heterotopic pregnancies occurred during the study, 14 in the ET group, and 33 in the FET group. The total heterotopic pregnancy rate was 0.29%. The heterotopic pregnancy rate in the ET group was 0.37% and 0.27% in the FET group. We found that the ectopic and heterotopic pregnancy rates were lower in the FET group than in the ET group, but there was no statistical difference (Table 2).

To clarify whether the type of embryo transfer (ET or FET) was a risk factor for EP after IVF–ET, relevant maternal characteristics and embryo transfer procedure characteristics were used as confounders. The logistic regression results showed no statistical difference in EP risk between the FET and ET groups (crude OR 0.96 (0.75–1.24), *p* > 0.05; adjusted OR 0.93 (0.71–1.22), *p* > 0.05). Similarly, there was no statistical difference in heterotopic pregnancy risk between the FET and ET groups (crude OR 0.73 (0.39–1.37), *p* > 0.05; adjusted OR 0.72 (0.34–1.50), *p* > 0.05) (Table 3).

### 3.3. Ectopic Pregnancy Outcome between Different Ovarian Stimulation Groups in ET Cycles

Moreover, we explored whether different ovarian stimulation protocols in ET cycles had an impact on EP rate. Our results showed that 3803 fresh ET cycles included 1535 GnRH-ant protocols, 578 prolonged agonist protocols, 490 short agonist protocols, and 1200 long agonist protocols. The total EP rate was 2.16% (82/3803), 3.32% (51/1535) in the GnRH-ant protocol group, 1.90% (11/578) in the prolonged agonist protocol group, 1.84% (9/490) in the short agonist protocol group, and 0.92% (11/1200) in the long agonist protocol group. The EP rate in the long agonist protocol group was significantly lower than that in the GnRH-ant protocol group (0.92% vs. 3.32%, *p* < 0.05). No difference in EP rate was noted between all groups. The total heterotopic pregnancy rate was 0.37% (14/3803), 0.39% (6/1535) in the GnRH-ant protocol group, 0.17% (1/578) in the prolonged agonist protocol group, 0.41% (2/490) in the short agonist protocol group, and 0.42% (5/1200) in the long agonist protocol group. No difference in heterotopic pregnancy rate was noted between all groups (Table 4).

The logistic regression results showed that, after adjustment for EP history and other covariates, the OR for EP (vs. GnRH-ant protocol group) was 0.59 (0.30–1.16) for the prolonged agonist protocol, 0.81 (0.38–1.71) for the short agonist protocol, and 0.45 (0.22–0.93) for the long agonist protocol. The long agonist protocol in ET cycles had a significantly lower risk of EP than the GnRH-ant protocol. The OR for heterotopic pregnancy rate (vs. the GnRH-ant protocol group) was 0.43 (0.05–3.58) for the prolonged agonist protocol, 1.05 (0.19–5.84) for the short agonist protocol, and 1.00 (0.26–3.92) for the long agonist protocol. The risk of heterotopic pregnancy in different ovarian stimulation protocols was comparable (Table 5).

## 4. Discussion

The first pregnancy reported after IVF–ET was a tubal pregnancy [21]. In recent years, significant advances have been made in assisted reproductive technology, but EP remains an important complication of IVF–ET. The EP rate after IVF–ET was 2.09% in our center from January 2013 to March 2022, which was consistent with global data [22]. In assisted reproduction, the occurrence of EP is not only a waste of precious embryos, but may also lead to a loss of function of the affected fallopian tube. Although conservative treatment or surgical removal of the affected fallopian tube does not affect ovarian response [23], the incidence of EP in a second pregnancy is significantly higher [24]. Therefore, it is particularly important to reduce the incidence of EP in ART.

The prevalence of the freeze-all policy in reproductive medicine is rising globally. The question of whether FET is a protective factor for EP after IVF–ET has been controversial. Our study concluded that the EP rate was slightly lower in FET cycles than in fresh ET cycles, but there was no significant difference (2.16% vs. 2.07%, *p* > 0.05), and logistic regression results showed no statistical difference in EP risk between the FET and ET groups (adjusted OR 0.93 (0.71–1.22), *p* > 0.05). Our findings were consistent with many other studies. Decleer’s team found that the incidence of the EPs per established clinical pregnancy was 1.92% for the fresh ET cycles vs. 1.28% for the FET cycles, which had no significant difference in a large cohort of 11,831 patients [18]. Bu’s team found a similar conclusion in a 6-year, single-center study of 18,432 pregnancies [16].

Compared to previous studies, our study included risk factors associated with EP after IVF–ET to adjust the final outcome. In particular, the inclusion of history of EP as an important factor made the results more accurate. In some previous studies with large samples, Santos’s team analyzed 161,967 pregnancies from database of the Human Fertilisation and Embryology Authority, and came to the same conclusion that we reached [25]. Perkins’s team found EP rate in FET was lower than that in ET after analyzing 553,577 pregnancies from the database of the National Assisted Reproductive Technology Surveillance System [26]. However, well-established risk factors of EP, such as history of EP, could not be assessed due to the use of these particular data. Our study took this problem into account, and improved the accuracy of the findings. Tubal infertility is the main factor impacting EP rate in IVF–ET cycles [16], and an infertility disease database (IDBB) may help us to decipher the mechanisms at play between tubal infertility and EP in the future [27].

Compared to fresh ET cycles, the hormone levels of patients in FET cycles are closer to physiological conditions. Controlled hyperovulation in the fresh ET cycles may increase the junctional zone contractility to expel the embryo from the uterine cavity [28]. The endometrium of the natural cycle is most tolerant during the implantation window. Controlled hyperovulation may cause the advancement of endometrial maturation [29] and reduce the endometrial tolerance, which may cause embryo–endometrium asynchrony, affecting the implantation [30]. Hormone levels in the supraphysiological state during controlled hyperovulation may also affect the fallopian tubes by increasing the expression of inflammatory factors and affecting ciliary-beat frequency [31,32]. Mock embryo transfer has shown that 38% of the transfer fluid may back up into the fallopian tube [33]. Thus, altered hormone levels during controlled hyperovulation may increase uterine contractions, prevent normal endometrial implantation of the transferred embryo and cause impaired tubal peristalsis or an abnormal expression of inflammatory factors, all of which can lead to EP.

We further analyzed the EP rate between different ovarian stimulation groups in ET cycles. The EP rate and risk in long agonist protocol group were significantly lower than that in the GnRH-ant protocol group (0.92% vs. 3.32%, *p* < 0.05) (OR 0.45 (0.22–0.93), *p* < 0.05). Our findings were consistent with Laura’s results, which showed the odds of EP observed in GnRH antagonist cycles were higher than in the GnRH agonist flare cycles (OR 1.19 (1.04–1.37), *p* < 0.001) [34]. In addition, Weiss’s team found a 5.53% EP rate under the GnRH-ant protocol [35]. However, little is known about the effects of extrapituitary GnRH on early implantation processes. Alternatively, the immune response system may play an important role in the interaction between embryonic and maternal tissues [34].

Our study had some limitations. First, our study was a retrospective observational study that assigned patients to different treatment groups based on clinical practice, which lead to selection bias. Therefore, further prospective cohort studies should be conducted. Second, our clinical data were all from one reproductive center. Therefore, further research on this topic is needed to gain support from multiple reproductive centers.

Our study demonstrated a slightly lower rate of EP in FET cycles than in fresh ET cycles, but there was no significant difference. The long agonist protocol in ET cycles had a significantly lower risk of EP than the GnRH-ant protocol. Due to the large sample size, there was a statistically significant difference between the two groups in terms of maternal and cycle characteristics of the patients, but the actual difference was small, and we performed logistic regression analysis of all these characteristics as confounders to adjust the results.

## Figures and Tables

**Figure 1 jcm-11-03386-f001:**
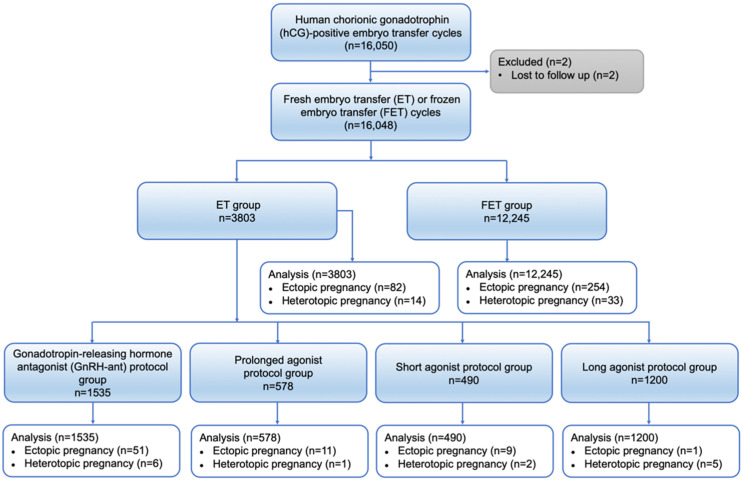
Flow chart of cohort screening in the study.

**Table 1 jcm-11-03386-t001:** Patient demographics and cycle characteristics.

Characteristics	ET Group (*n* = 3803)	FET Group (*n* = 12,245)	*p* Value
Age of women (year)	39.45 ± 5.999	35.19 ± 4.531	<0.001
BMI of women	21.74 ± 3.057	22.27 ± 3.367	<0.001
Infertility duration(year)	3.60 ± 2.996	3.07 ± 2.696	<0.001
Infertility causes, *n* (%)			
Tubal factor	2364 (62.16%)	7058 (57.64%)	<0.001
Male factor	551 (14.49%)	1587 (12.95%)	<0.05
Endometriosis	155 (4.08%)	412 (3.36%)	<0.05
PCOS	107 (2.81%)	849 (6.93%)	<0.001
Uterine factor	51 (1.34%)	206 (1.68%)	0.14
DOR	48 (1.26%)	193 (1.58%)	0.16
Others	527 (13.86%)	1940 (15.84%)	<0.01
History of ectopic pregnancy, *n* (%)	277 (7.28%)	1818 (14.85%)	<0.001
Nature cycle, *n* (%)			
Yes	0 (0.00%)	1281 (10.46%)	<0.001
No	3803 (100.00%)	10,964 (89.54%)	<0.001
No. of embryos transferred, *n* (%)			
1	694 (18.25%)	4111 (33.57%)	<0.001
2	2883 (75.81%)	8132 (66.41%)	<0.001
3	224 (5.89%)	2 (0.02%)	<0.001
4	2 (0.05%)	0 (0.00%)	<0.05
Stage of embryos transferred, *n* (%)			
Blastocyst	27 (0.71%)	3593 (29.34%)	<0.001
Non-blastocyst	3776 (99.29%)	8741 (71.38%)	<0.001
Endometrial thickness (mm)	11.54 ± 2.720	10.83 ± 2.434	<0.001

Abbreviations: ET= fresh embryo transfer; FET=frozen embryo transfer; BMI = body mass index; NS = no significant difference; PCOS = polycystic ovary syndrome; DOR = Diminished ovarian reserve.

**Table 2 jcm-11-03386-t002:** Ectopic pregnancy rate between ET and FET groups.

	ET Group (*n* = 3803)	FET Group (*n* = 12,245)	*p* Value
Ectopic pregnancy, *n* (%)	82 (2.16%)	254 (2.07%)	0.76
Heterotopic pregnancy, *n* (%)	14 (0.37%)	33 (0.27%)	0.33

**Table 3 jcm-11-03386-t003:** Crude and adjusted odds ratios (OR) of ectopic pregnancy and heterotopic pregnancy between ET and FET groups (all *p* > 0.05).

	Ectopic Pregnancy		Heterotopic Pregnancy	
	Crude OR (95% CI)	Adjusted OR (95% CI)	Crude OR (95% CI)	Adjusted OR (95% CI)
ET group	Reference	Reference	Reference	Reference
FET group	0.96 (0.75–1.24)	0.93 (0.71–1.22)	0.73 (0.39–1.37)	0.72 (0.34–1.50)

**Table 4 jcm-11-03386-t004:** Ectopic pregnancy rate between different ovarian stimulation groups in ET cycles (^a^: long agonist protocol compared with GnRH-ant protocol, *p* < 0.05).

	GnRH-ant Protocol (*n* = 1535)	Prolonged Agonist Protocol (*n* = 578)	Short Agonist Protocol (*n* = 490)	Long Agonist Protocol (*n* = 1200)
Ectopic pregnancy, *n* (%)	51 (3.32%)	11 (1.90%)	9 (1.84%)	11 (0.92%) ^a^
Heterotopic pregnancy, *n* (%)	6 (0.39%)	1 (0.17%)	2 (0.41%)	5 (0.42%)

**Table 5 jcm-11-03386-t005:** Crude and adjusted odds ratios (OR) of ectopic pregnancy and heterotopic pregnancy between different ovarian stimulation groups in ET cycles (* *p* < 0.05, *** *p* < 0.001).

	Ectopic Pregnancy		Heterotopic Pregnancy	
	Crude OR (95% CI)	Adjusted OR (95% CI)	Crude OR (95% CI)	Adjusted OR (95% CI)
GnRH-ant protocol	Reference	Reference	Reference	Reference
prolonged agonist protocol	0.57 (0.29–1.09)	0.59 (0.30–1.16)	0.44 (0.05–3.68)	0.43 (0.05–3.58)
short agonist protocol	0.54 (0.27–1.11)	0.81 (0.38–1.71)	1.04 (0.21–5.19)	1.05 (0.19–5.84)
long agonist protocol	0.27 (0.14–0.52) ***	0.45 (0.22–0.93) *	1.07 (0.33–3.50)	1.00 (0.26–3.92)

## Data Availability

The data presented in this study are available on request from the corresponding author.

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
