# Peer review of "Differences in Ectopic Pregnancy Rates between Fresh and Frozen Embryo Transfer after In Vitro Fertilization: A Large Retrospective Study"

_jcm, 2022, doi:10.3390/jcm11123386_

Round 1

Reviewer 1 Report

1. Line 36-37, please provide information and/or data on the reasons fresh vs frozen may be associated with EP. What is the perceived association or risk factor? You want to strengthen your argument that this is not just related to physiologic factors of needing ART in the first place.

2. Line 49, "ultimately" might make more sense then moreover. 

3. Line 59, do not give away the results in the introduction. Instead, would include hypothesis here.

4. Typically in materials & methods, we include the #s in the results, not in the methods (ie., you would not give n=##). Move to Results. Line 65-75.

5. Line 72 and in Figure 1 and line 81 "prolonged agonist protocol group" not "rolonged agonist..."

6. Ovarian stimulation & oocyte retrieval operation could be collapsed into 1 paragraph. 

7. Line 142 "for categorical variables..." - this is not needed, it is obvious. 

8. Line 148, say "the regression model adjusted for ..." instead of "took into possible confounding factors." Unless you did not adjust for these factors, for which you should state explicitly. 

9. Your populations are widely different in terms of BMI, age, etc. Please comment on why you think this is. Also, worth commenting on why there are 4x the number of FET cycles, which is fine, just comment on why that is. Why did women with higher % of ectopic elect for FET?

10. Classically, still provide p values even if not statistically significant. 

11. Line 162, capitalize Table 1. 

12. Line 169 - which % is correct with regards to the ET group, 2.09 or 2.16? Same with heterotypic on Line 171-172.

13. Table 2, provide p values even if NS.

14. Good presentation of conclusion in first 3 paragraphs. 

15. Laura's results needs a citation, in Line 268.

16. Did you address the paper's limitations? Would make a separate paragraph for this, and then end on paper's strengths. 

Author Response

We thank the reviewer for your valuable comments. We have revised our paper as requested, and our responses are demonstrated point-by-point as follows.

Point 1: Line 36-37, please provide information and/or data on the reasons fresh vs frozen may be associated with EP. What is the perceived association or risk factor? You want to strengthen your argument that this is not just related to physiologic factors of needing ART in the first place.

Response 1: Thank you for your comment. Previous studies have shown that EP was related with high level of estrogen, which led to increased uterine contractions(PMID: 17623538ï¼›PMID: 15521862), and also related with high level of progesterone leading to ciliary dysfunction during assisted reproductive technology(ART)( PMID: 10806589). Osamu Ishihara‘s (PMID: 25241365) and Huang’s (PMID: 21377154) studies both showed that the higher EP rate after fresh ET was related to the negative effect of ovarian stimulation on endometrial receptivity, but the majority of frozen-thawed ETs used natural cycle preparation. We have added the above information to the article, and emphasized the relevance of the type of embryo transferred (fresh or frozen embryo transfer) and EP. ( line35-53).

Point 2: Line 49, "ultimately" might make more sense then moreover. 

Response 2: Thank you for your comment. We used “ultimately” in place of “moreover”.( line59)

Point 3: Line 59, do not give away the results in the introduction. Instead, would include hypothesis here.

Response 3: Thank you for your comment. We deleted the results in the introduction and gave the hypothesis that the EP rate might be associated with the type of embryo transferred (fresh or frozen embryo transfer). ( line69-70)

Point 4: Typically in materials & methods, we include the #s in the results, not in the methods (ie., you would not give n=##). Move to Results. Line 65-75.

Response 4: Thank you for your comment. We removed detailed #s in the section of Results. ( line78-81)

Point 5: Line 72 and in Figure 1 and line 81 "prolonged agonist protocol group" not "rolonged agonist..."

Response 5: Thank you for pointing out that. We used “prolonged agonist protocol group” in place of “rolonged agonist protocol group”.

Point 6: Ovarian stimulation & oocyte retrieval operation could be collapsed into 1 paragraph. 

Response 6: Thank you for your comment. Ovarian stimulation & oocyte retrieval operation was collapsed into 1 paragraph. ( line169-190)

Point 7: Line 142 "for categorical variables..." - this is not needed, it is obvious. 

Response 7: Thank you for your suggestion. We deleted the sentence "for categorical variables...". ( line244)

Point 8: Line 148, say "the regression model adjusted for ..." instead of "took into possible confounding factors." Unless you did not adjust for these factors, for which you should state explicitly. 

Response 8: Thank you for your comment. We used “the regression model adjusted for ...” in place of “took into possible confounding factors”.( line249-252)

Point 9: Your populations are widely different in terms of BMI, age, etc. Please comment on why you think this is. Also, worth commenting on why there are 4x the number of FET cycles, which is fine, just comment on why that is. Why did women with higher % of ectopic elect for FET?

Response 9: Thank you for your comment. The wide difference in BMI, age, etc might due to large sample size of the study. We could see that there were a lot of statistically significant differences, which were actually small, concerning patient demographics and cycle characteristics. Therefore, the highly statistical difference may be contributed to large sample size, not meaning clinical relevance.

With the development of vitrification freezing technology, freeze-all policy has been widely used in reproductive medicine. It has been suggested that by performing freeze-all policy, embryos are transferred into the uterus in a more physiologic intrauterine environment, which avoids asynchrony between endometrium and embryos caused by supra-physiologic hormonal levels during controlled ovarian stimulation (PMID: 30689865ï¼›PMID: 30388233) . Therefore, in our center, physicians prefer to perform FET to avioid OHSS and increase the liver birth, which is the reason why women with higher % of ectopic elect for FET in this study.

Point 10: Classically, still provide p values even if not statistically significant. 

Response 10: Thank you for your comment. We provided all p values in the article instead of NS. (Table1, Table2)

Point 11: Line 162, capitalize Table 1. 

Response 11: Thank you for your comment. We capitalized Table 1. ( line263)

Point 12: Line 169 - which % is correct with regards to the ET group, 2.09 or 2.16? Same with heterotypic on Line 171-172. 

Response 12: Thank you for your comment. The total ectopic pregnancy rate in this study was 2.09%. The ectopic pregnancy rate in the ET group was 2.16%. The total heterotopic pregnancy rate in this study was 0.29%. The heterotopic pregnancy rate in the ET group was 0.37%. We have rewritten the relevant sentences to make it clearer. ( line287-290)

Point 13: Table 2, provide p values even if NS.

Response 13: Thank you for your comment. We provided all p values in the article instead of NS (Table 2).

Point 14: Good presentation of conclusion in first 3 paragraphs. 

Response 14: Thank you for your kind and positive evaluation.

Point 15: Laura's results needs a citation, in Line 268. 

Response 15: Thank you for your comment. We cited Laura's results in the article. ( line398)

Point 16: Did you address the paper's limitations? Would make a separate paragraph for this, and then end on paper's strengths. 

Response 16: Thank you for your suggestion. We addressed the study's limitations in Disscusion. First, our study was a retrospective observational study that assigned patients to different treatment groups based on clinical practice, which lead to selection bias. Therefore, further prospective cohort studies should be conducted. Second, our clinical data were all from one reproductive centre. Therefore, further research on this topic is needed to gain support from multiple reproductive centres. We also emphasized paper's strengths at the end of the paper.( line438-442)

Reviewer 2 Report

I congratulate you for the work and time you dedicated for this study. The results of your study seem to be similar to previous studies and articles, so I consider the originality and novelty of your findings to be average. Nonetheless I appreciate that this is a well-documented study with a large cohort of patients. Also, although you did not find a difference in ectopic pregnancy rates between fresh and frozen embryo transfer after In-Vitro fertilization, this is still considered an important finding.

Author Response

We thank the reviewer for your valuable comments. We have revised our paper as requested, and our responses are demonstrated point-by-point as follows.

Point 1: I congratulate you for the work and time you dedicated for this study. The results of your study seem to be similar to previous studies and articles, so I consider the originality and novelty of your findings to be average. Nonetheless I appreciate that this is a well-documented study with a large cohort of patients. Also, although you did not find a difference in ectopic pregnancy rates between fresh and frozen embryo transfer after In-Vitro fertilization, this is still considered an important finding.

Our Response : Thank you for your kind and positive evaluation. Compared to previous studies, our study included risk factors associated with ectopic pregnancy after IVF-ET, such as history of ectopic pregnancy which was not included in most previous studies, to ensure the accuracy of the results. In addition, we futher analyzed EP rates in different ovarian stimulation protocols in ET cycles to ensure the generalizability of the results. Our study was a retrospective observational study that assigned patients to different treatment groups based on clinical practice, which might lead to selection bias, thus limiting the work. Therefore, further prospective cohort studies should be conducted in more homogenised patients groups to test and verify the conclusions.